# Detection of significant antiviral drug effects on COVID-19 with reasonable sample sizes in randomized controlled trials: A modeling study

Shoya Iwanami[1,2☯], Keisuke Ejima[3☯]*, Kwang Su Kim[1,2], Koji Noshita[1], Yasuhisa Fujita[1,2], Taiga Miyazaki[4], Shigeru Kohno[5], Yoshitsugu Miyazaki[6], Shimpei Morimoto[7], Shinji Nakaoka[8], Yoshiki Koizumi[9], Yusuke Asai[10], Kazuyuki Aihara[11], Koichi Watashi[12,13,14], Robin N. Thompson[15,16], Kenji Shibuya[17], Katsuhito Fujiu[18,19], Alan S. Perelson[20,21☯], Shingo Iwami[1,2,22,23,24,25☯]*, Takaji Wakita[12]

1 Department of Biology, Faculty of Sciences, Kyushu University, Fukuoka, Japan, 2 interdisciplinary Biology Laboratory (iBLab), Division of Biological Science, Graduate School of Science, Nagoya University, Nagoya, Japan, 3 Department of Epidemiology and Biostatistics, Indiana University School of Public Health-Bloomington, Indiana, United States of America, 4 Department of Infectious Diseases, Nagasaki University Graduate School of Biomedical Sciences, Nagasaki, Japan, 5 Nagasaki University, Nagasaki, Japan, 6 Department of Chemotherapy & Mycoses and Leprosy Research Center, National Institute of Infectious Diseases, Tokyo, Japan, 7 Institute of Biomedical Sciences, Nagasaki University, Japan, 8 Faculty of Advanced Life Science, Hokkaido University, Sapporo, Japan, 9 National Center for Global Health and Medicine, Tokyo, Japan, 10 Disease Control and Prevention Center, National Center for Global Health and Medicine, Tokyo, Japan, 11 International Research Center for Neurointelligence, The University of Tokyo Institutes for Advanced Study, The University of Tokyo, Tokyo, Japan, 12 Department of Virology II, National Institute of Infectious Diseases, Tokyo, Japan, 13 Department of Applied Biological Science, Tokyo University of Science, Noda, Japan, 14 Institute for Frontier Life and Medical Sciences, Kyoto University, Kyoto, Japan, 15 Mathematics Institute, University of Warwick, Coventry, United Kingdom, 16 Zeeman Institute for Systems Biology and Infectious Disease Epidemiology Research, University of Warwick, Coventry, United Kingdom, 17 Institute for Population Health, King's College London, London, United Kingdom, 18 Department of Cardiovascular Medicine, Graduate School of Medicine, The University of Tokyo, Tokyo, Japan, 19 Department of Advanced Cardiology, Graduate School of Medicine, The University of Tokyo, Tokyo, Japan, 20 Theoretical Biology and Biophysics Group, Los Alamos National Laboratory, Los Alamos, New Mexico, United States of America, 21 New Mexico Consortium, Los Alamos, New Mexico, United States of America, 22 Institute of Mathematics for Industry, Kyushu University, Fukuoka, Japan, 23 Institute for the Advanced Study of Human Biology (ASHBi), Kyoto University, Kyoto, Japan, 24 NEXT-Ganken Program, Japanese Foundation for Cancer Research (JFCR), Tokyo, Japan, 25 Science Groove Inc., Fukuoka, Japan

☯ These authors contributed equally to this work.
* kejima@iu.edu (KE); iwami.iblab@bio.nagoya-u.ac.jp, iwamishingo@gmail.com (SI)

## Abstract

### Background

Development of an effective antiviral drug for Coronavirus Disease 2019 (COVID-19) is a global health priority. Although several candidate drugs have been identified through in vitro and in vivo models, consistent and compelling evidence from clinical studies is limited. The lack of evidence from clinical trials may stem in part from the imperfect design of the trials. We investigated how clinical trials for antivirals need to be designed, especially focusing on the sample size in randomized controlled trials.

**Data Availability Statement:** All relevant data are within the manuscript and its Supporting information files.

**Funding:** This study was supported by Japan Society for the Promotion of Science (JSPS) KAKENHI Grant Numbers JP19J12319 (to S. Iwanami), JP18KT0018 (to S.I.), JP18H01139 (to S.I.), JP16H04845 (to S.I.), JP17H04085 (to K.W.), JP18K18146 (to K.E.), JP20H05042 (to S.I.), JP19H04839 (to S.I.), JP18H05103 (to S.I.); Japan Agency for Medical Research and Development (AMED) Grant Numbers JP19gm1310002 (to S.I.), JP20wm0325007h0001 (to S.I.), JP20wm0325004s0201 (to S.I.), JP20wm0325012s0301 (to S.I.), JP20wm0325015s0301 (to S.I.), JP19fk0410023s0101 (to S.I.), JP19fk0108050h0003 (to S.I.), JP19fk0108156h0001 (to S.I.), JP20fk0108140s0801 (to S.I.), JP20fk0108413s0301 (to S.I.), JP19fk0210036h0502 (to S.I.), JP19fk0210036j0002 (to K.W.), JP19fk0310114h0103 (to S.I.), JP19fk0310114j0003 (to K.W.), JP19fk0310101j1003 (to K.W.), JP19fk0310103j0203 (to K.W.); Japan Science and Technology Agency (JST) MIRAI (to S.I. and K.W.); Moonshot R&D Grant Number JPMJMS2021 (to K.A. and S.I.) and JPMJMS2025 (to S.I.); Mitsui Life Social Welfare Foundation (to S.I. and K.W.); Shin-Nihon of Advanced Medical Research (to S. I.); Suzuken Memorial Foundation (to S.I.); Life Science Foundation of Japan (to S.I.); SECOM Science and Technology Foundation (to S.I.); The Japan Prize Foundation (to S.I.); Daiwa Securities Health Foundation (to S.I.); The Yasuda Medical Foundation (to K.W.); Smoking Research Foundation (to K.W.); and The Takeda Science Foundation (to K.W.); NSF PHY-2031756 (to A.S. P.); NIH R01-OD011095 and R01-AI028433 (to A. S.P.); and Los Alamos National Laboratory LDRD Program (to A.S.P.); the MIDAS Coordination Center (MIDASSUGP2020-6) by a grant from the NIGMS (3U24GM132013-02S2) (to K.E.); Meiji Yasuda Life Foundation of Health and Welfare (K. E.). The funders had no role in study design, data collection and analysis, decision to publish, or preparation of the manuscript.

**Competing interests:** The authors have declared that no competing interests exist.

**Abbreviations:** ACE2, angiotensin converting enzyme 2; AUC, area under the curve; BIC, Bayesian information criteria; BICc, corrected Bayesian information criteria; COVID-19, Coronavirus Disease 2019; Ct, cycle threshold;

## Methods and findings

A modeling study was conducted to help understand the reasons behind inconsistent clinical trial findings and to design better clinical trials. We first analyzed longitudinal viral load data for Severe Acute Respiratory Syndrome Coronavirus 2 (SARS-CoV-2) without antiviral treatment by use of a within-host virus dynamics model. The fitted viral load was categorized into 3 different groups by a clustering approach. Comparison of the estimated parameters showed that the 3 distinct groups were characterized by different virus decay rates ($p$-value < 0.001). The mean decay rates were 1.17 $d^{-1}$ (95% CI: 1.06 to 1.27 $d^{-1}$), 0.777 $d^{-1}$ (0.716 to 0.838 $d^{-1}$), and 0.450 $d^{-1}$ (0.378 to 0.522 $d^{-1}$) for the 3 groups, respectively. Such heterogeneity in virus dynamics could be a confounding variable if it is associated with treatment allocation in compassionate use programs (i.e., observational studies).

Subsequently, we mimicked randomized controlled trials of antivirals by simulation. An antiviral effect causing a 95% to 99% reduction in viral replication was added to the model. To be realistic, we assumed that randomization and treatment are initiated with some time lag after symptom onset. Using the duration of virus shedding as an outcome, the sample size to detect a statistically significant mean difference between the treatment and placebo groups (1:1 allocation) was 13,603 and 11,670 (when the antiviral effect was 95% and 99%, respectively) per group if all patients are enrolled regardless of timing of randomization. The sample size was reduced to 584 and 458 (when the antiviral effect was 95% and 99%, respectively) if only patients who are treated within 1 day of symptom onset are enrolled. We confirmed the sample size was similarly reduced when using cumulative viral load in log scale as an outcome.

We used a conventional virus dynamics model, which may not fully reflect the detailed mechanisms of viral dynamics of SARS-CoV-2. The model needs to be calibrated in terms of both parameter settings and model structure, which would yield more reliable sample size calculation.

## Conclusions

In this study, we found that estimated association in observational studies can be biased due to large heterogeneity in viral dynamics among infected individuals, and statistically significant effect in randomized controlled trials may be difficult to be detected due to small sample size. The sample size can be dramatically reduced by recruiting patients immediately after developing symptoms. We believe this is the first study investigated the study design of clinical trials for antiviral treatment using the viral dynamics model.

### Author summary

#### Why was this study done?

- Most clinical studies of antiviral drugs for Severe Acute Respiratory Syndrome Coronavirus 2 (SARS-CoV-2) have failed to observe a statistically significant effect, which may be due to poor designs of clinical trials.

FDA, Food and Drug Administration; SARS-CoV-2,
Severe Acute Respiratory Syndrome Coronavirus
2.

- It is not well studied how clinical trials for antiviral drugs should be designed. Especially, sample size calculation methodology needs to be established.

### What did the researchers do and find?

- SARS-CoV-2 virus dynamics was quantified by fitting a virus dynamic model to longitudinal viral load data.

- Cluster analysis of the fitted viral loads revealed 3 distinct groups characterized by different virus decay rates, which could be a confounding factor in observational studies.

- Simulation mimicking randomized controlled trials demonstrated that sample size would be unreasonably large (>11,000 per group) if the timing of treatment initiation is not considered. The sample size is significantly reduced by including only patients enrolled early after symptom onset.

### What do these findings mean?

- Randomized controlled trials for antiviral drugs should recruit patients as early as possible after symptom onset or set inclusion criteria based on the time since symptom onset to observe statistically significant results.

- More precise models reflecting the features of SARS-CoV-2 infection may provide more reliable sample size estimates.

## Introduction

Development of an effective antiviral drug for Coronavirus Disease 2019 (COVID-19) is a global health priority. Along with the development of new antiviral drugs, repurposing of existing drugs for COVID-19 treatment has accelerated [1–9]. Some antiviral drugs have shown high efficacy against Severe Acute Respiratory Syndrome Coronavirus 2 (SARS-CoV-2) in both in vitro and in vivo models [10,11]. A number of clinical studies such as compassionate use programs and clinical trials have been conducted or are underway to test the efficacy of Food and Drug Administration (FDA)-approved drugs, such as lopinavir/ritonavir, chloroquine, favipiravir, and remdesivir [12–17]. Different drugs have different modes of action, but the majority of the candidate antiviral drugs for SARS-CoV-2 are expected to block virus replication. Lopinavir/ritonavir are HIV protease inhibitors, and remdesivir was originally developed to mitigate the replication of hepatitis C viruses (and considered potentially useful for Ebola virus). Other nucleoside analogues [18,19] are also candidates for mitigating SARS-CoV-2 replication within the host.

However, the results from those clinical studies were often nonsignificant, and, sometimes, inconsistent. This may be in part attributable to a nonrigorous study design, which masks the true efficacy of antivirals [20]. Clinical trial design usually takes months to formulate the study protocol (i.e., dose of drugs, clinical outcomes to be evaluated, sample size, and assessment of safety) and requires collecting preliminary data. However, the urgent need to find effective antiviral treatments for COVID-19 may have led to rushed studies.

In compassionate use programs (i.e., observational studies), whether and when antiviral treatment is initiated is determined by health practitioners along with patients and their next

of kin. By the very nature of these studies, potential confounders, such as the patients' clinical characteristics and preexisting conditions, influence both treatment–control allocation and clinical outcomes. As a consequence, conclusions from the program could be biased even when all observable confounders are addressed in the analysis [21]. However, such programs are widely used for hypothesis building. Contrary to compassionate use programs, clinical trials, particularly randomized controlled trials, are considered robust against confounder effects and the most reliable study design. S1 Table summarizes the current major clinical studies for antiviral treatment of SARS-CoV-2 (as of May 22, 2020). Indeed, the results from these clinical studies have yielded null or inconsistent findings. For example, compassionate use of hydroxy-chloroquine was reported in many articles, but the findings were not consistent. Gautret and colleagues reported significant antiviral efficacy [22], whereas Geleris and colleagues could not replicate the result [23].

To help understand the mechanism behind the inconsistent findings, we parametrized the virus dynamics model that we previously developed [24–26] by using longitudinal viral load data extracted from clinical studies and further ran simulations by adding antiviral effects to the model. Here, we demonstrate that at least 2 factors can mask the effects of antiviral drugs in clinical studies for COVID-19: (1) heterogeneity in virus dynamics among patients; and (2) late timing of treatment initiation. We also propose a novel approach, to the best of our knowledge, to calculating the sample size (i.e., the required or minimum sample size needed to infer whether the antiviral drug is effective assuming the drug is truly effective) accounting for within-host virus dynamics.

## Methods

### Study data

The longitudinal viral load data examined in our study were extracted from the published studies of SARS-CoV-2: Young and colleagues [27], Zou and colleagues [28], Kim and colleagues [29], and Wölfel and colleagues [30]. For consistency, the viral load data measured from upper respiratory specimens were used. We excluded patients who received antiviral treatment and for whom data were measured on only 1 or 2 days (because 1 or 2 data points are not enough to estimate parameters). We converted cycle threshold (Ct) values to viral RNA copy number values, where these quantities are inversely proportional to each other [28]. In total, we use the data from 30 patients. To extract the data from the images in those papers, we used the software DataThief III (version 1.5, Bas Tummers, www.datathief.org).

### Mathematical model for virus dynamics without and with antiviral treatment

SARS-CoV-2 virus dynamics without antiviral treatment is described by a mathematical model previously proposed in [24,31,32]:

$$\frac{df(t)}{dt} = -\beta f(t)V(t), \tag{1}$$

$$\frac{dV(t)}{dt} = \gamma f(t)V(t) - \delta V(t), \tag{2}$$

where $f(t)$ is the relative fraction of uninfected target cells at time $t$ to those at time 0, and $V(t)$ is the amount of virus at time $t$, respectively. Both $f(t)$ and $V(t)$ are in linear scale. The parameters $\beta$, $\gamma$, and $\delta$ represent the rate constant for virus infection, the maximum rate constant for viral replication, and the per capita death rate of virus-producing cells, respectively. Note that

$\delta$ implicitly includes the effects of the immune response in killing infected cells, e.g., by cytotoxic T lymphocytes. All viral load data were fit using a nonlinear mixed-effect modeling approach, which estimates population parameters while accounting for interindividual variation in virus dynamics (see the next section for details). The day from symptom onset was used as a timescale (i.e., $t = 0$ at symptom onset).

The virus dynamic model under antiviral treatment (which we assume blocks virus replication) initiated at $t^*$ days after symptom onset can be described based on the above model as follows:

$$\frac{df(t)}{dt} = -\beta f(t)V(t), \tag{3}$$

$$\frac{dV(t)}{dt} = (1 - \varepsilon \times H(t))\gamma f(t)V(t) - \delta V(t), \tag{4}$$

where $H(t)$ is a Heaviside function indicating off and on treatment, defined as $H(t) = 0$ if $t < t^*$ (i.e., before treatment initiation); otherwise, $H(t) = 1$. $\varepsilon$ is the fraction of virus production inhibited by the therapy ($0 < \varepsilon \leq 1$). $\varepsilon = 1$ when the virus replication from the infected cells is totally inhibited (i.e., the antiviral effect is 100%). We evaluated the expected antiviral effect of the treatment on the outcomes (duration of virus shedding and cumulative viral load measured on a log scale) under different inhibition rates ($\varepsilon$) and initiation times ($t^*$). The effect of drugs that blocking de novo infection can be modeled by inhibiting both the $\beta f(t)V(t)$ and $\gamma f(t)$ $V(t)$ terms, and a drug promoting cytotoxicity can be modeled by increasing $\delta V(t)$, as we discussed in [31]. Unfortunately, because sufficient viral load data under antiviral drug therapy are not available yet, the antiviral effect ($\varepsilon$) of drugs in preclinical development and in clinical trials are still unknown. Therefore, we chose to examine hypothetical examples of drugs with 50%, 95%, or 99% efficacy. We used 50% and 99% efficacy to illustrate the difference in viral dynamics between patients with and without treatment (see section "SARS-CoV-2 virus dynamics and antiviral effect") and 95% and 99% efficacy in "Simulation mimicking a randomized controlled trial for antiviral drugs." Since clinical trials are performed only for drugs with sufficient efficacy (i.e., there is no reason to test drugs with weak efficacy), we believe this value range is reasonable.

## Parameter estimation with the nonlinear mixed-effects model

A nonlinear mixed-effects model was used to fit the viral dynamic model given by Eqs 1 and 2 to the longitudinal viral load data. The model included both a fixed effect (constant across patients) and a random effect (different between patients) in each parameter. Specifically, the parameter for patient $k$, $\vartheta_k (= \vartheta \times e^{\pi_k})$ is represented as a product of $\vartheta$ (a fixed effect) and $e^{\pi_k}$ (a random effect). $\pi_k$ follows the normal distribution with mean 0 and standard deviation $\Omega$. Fixed effects and random effects were estimated using the stochastic approximation expectation–maximization algorithm and empirical Bayes method, respectively. The conditional distribution of the vector of individual parameters was estimated for each patient using the Metropolis–Hastings algorithm and was used to calculate the 95% predictive interval of the viral load curve in Fig 1. The mixed-model approach is becoming more common in longitudinal viral load data analysis [26,33], because it can capture the heterogeneity in virus dynamics, and parameter estimation is feasible even for those with limited data. Fitting was performed using MONOLIX 2019R2 (www.lixoft.com) [34]. To account for data points under the detection limit (see the red dots in S1 Fig), the likelihood function reflected the likelihood that the

data are in the censoring interval (0 to the detection limit) given parameter values with a right-truncated Gaussian distribution [35].

## Clustering of individual viral load dynamics

As observed in Fig 1, the virus dynamics has huge heterogeneity between patients. For some patients, viral load declines rapidly, but for others, it persists for almost 1 month. It is ideal if the longitudinal viral load data can be directly compared between patients; however, the data collection intervals are not the same between patients, and the data under the detection limit are not quantifiable. Therefore, we used the fitted viral load every day since symptom onset, which is available from the best fit curve, for comparison. The fitted daily viral load values of each patient were rescaled by their maximum values and log-transformed. Then, hierarchical clustering was performed on the rescaled-transformed fitted daily viral load using the linkage function with Ward method [36] in SciPy [37]. Once multiple clusters are identified, estimated parameter distributions among the clusters were compared by ANOVA to assess the source of the difference in virus dynamics. Pairwise comparison was subsequently performed using Student $t$ test. The $p$-values of the pairwise Student $t$ test were adjusted by the Bonferroni correction.

## Simulation mimicking a randomized controlled trial for antiviral drugs

We mimicked randomized controlled trials using the model including the effects of an antiviral drug. The allocation ratio is assumed as 1:1 (control:treatment). A total of 20,000 parameter sets were randomly sampled from the estimated distributions of individual parameters. A longitudinal viral load time series for each individual was created based on their chosen parameter set. Note that for those in the treatment group, the antiviral effect ($\varepsilon$) was assumed to be constant. For sensitivity analysis, we used 2 different values of $\varepsilon$, 95% and 99%. To obtain a realistic simulation, treatment was initiated following the distribution of time from symptom onset to hospitalization obtained from Bi and colleagues: $lognorm$ (1.23, 0.79) (the mean is 4.64 days) [38], where the treatment was assumed to be initiated immediately after hospitalization. We also used a truncated distribution to mimic the randomized controlled trials including only patients recruited and treated early (within 0.5, 1, 2, 3, and 4 days after symptom onset).

We used 2 quantities as outcome measures: the duration of virus shedding from the onset of symptoms until the time the virus becomes undetectable ($T_D$) and the log10-transformed cumulative viral load, i.e., the area under the curve (AUC) of viral load ($\log_{10}(\text{AUC}) : \log_{10} \int_0^{T_D} V(s)ds$). Many clinical studies have used the duration of viral shedding as a primary outcome (see S1 Table), and previous theoretical studies have quantified the AUC. Both of the outcomes we use here are expected to be reduced under effective antiviral treatment.

From our simulations, we obtained 10,000 outcomes for each group (duration of virus shedding and cumulative viral load). The sample size was computed for different values of $\varepsilon$ using the 2-tailed Welch $t$ test with significance level and power as 0.05 and 80%, respectively.

## Results

### Heterogeneity in SARS-CoV-2 virus dynamics

SARS-CoV-2 viral load data were analyzed using a mathematical model to quantify the heterogeneity in virus dynamics among patients and to examine the source of the heterogeneity. Longitudinal viral load data from 30 patients from different countries were fitted simultaneously using a nonlinear mixed-effects modeling approach. With the estimated parameters for each patient (listed in S2 Table), viral loads since the time of symptom onset were fully

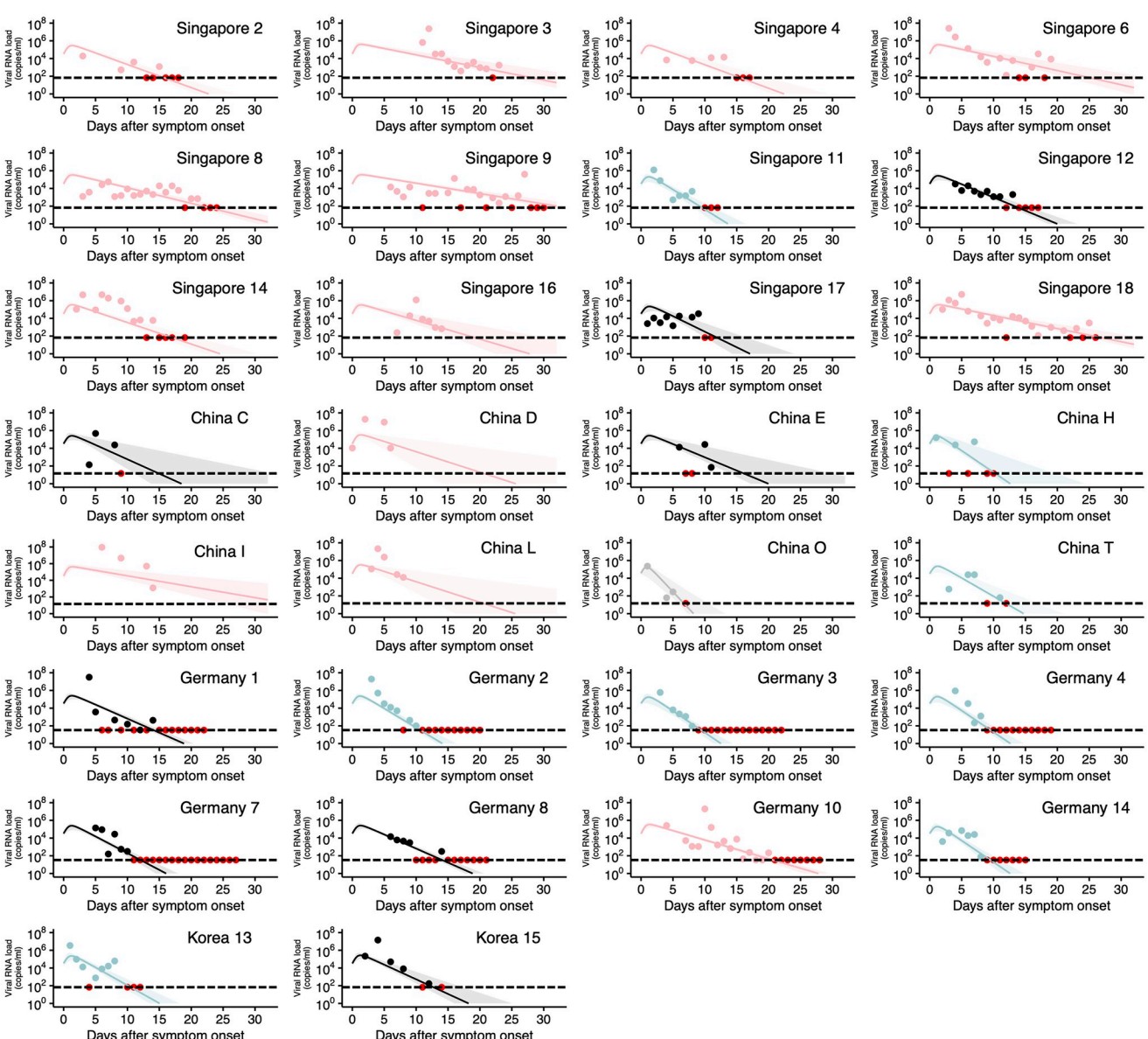

**Fig 1. Observed and fitted viral load data for individual patients.** Viral loads were measured using nasal swabs (China), pharyngeal swabs (Germany), nasopharyngeal swabs (Singapore and Korea), and oropharyngeal swab (Korea) for hospitalized SARS-CoV-2–infected patients. Note that the detection limits of the PCR assay for SARS-CoV-2 were 68.0 copies/ml (Singapore and Korea), 15.3 copies/ml (China), and 33.3 copies/ml (Germany), respectively, and are shown as dotted horizontal lines. The closed dots and curves correspond to the observed and the estimated viral load for each patient using their individual parameters given in S2 Table, respectively. Shaded regions correspond to 95% predictive intervals. Different colors of the dots and the lines (light blue, black, and pink) correspond to the 3 different types of patients characterized by rapid, medium, and slow viral load decay, respectively. The red dots represent the data at or under the detection limit regardless of the group. Patient IDs are the same as in the original papers if available. The underlying data for this figure can be found in S1 Data. SARS-CoV-2, Severe Acute Respiratory Syndrome Coronavirus 2.

reconstructed even when the viral load was missed or under the detection limit (Figs 1 and 2A). This reconstruction allowed us to quantitatively compare viral load dynamics between patients. The viral loads over time, which were reconstructed based on the mathematical model with the estimated parameters, were analyzed with a clustering approach (Fig 2B) and placed into 3 groups. Patient "China O" was detected as an outlier and was excluded from further analysis.

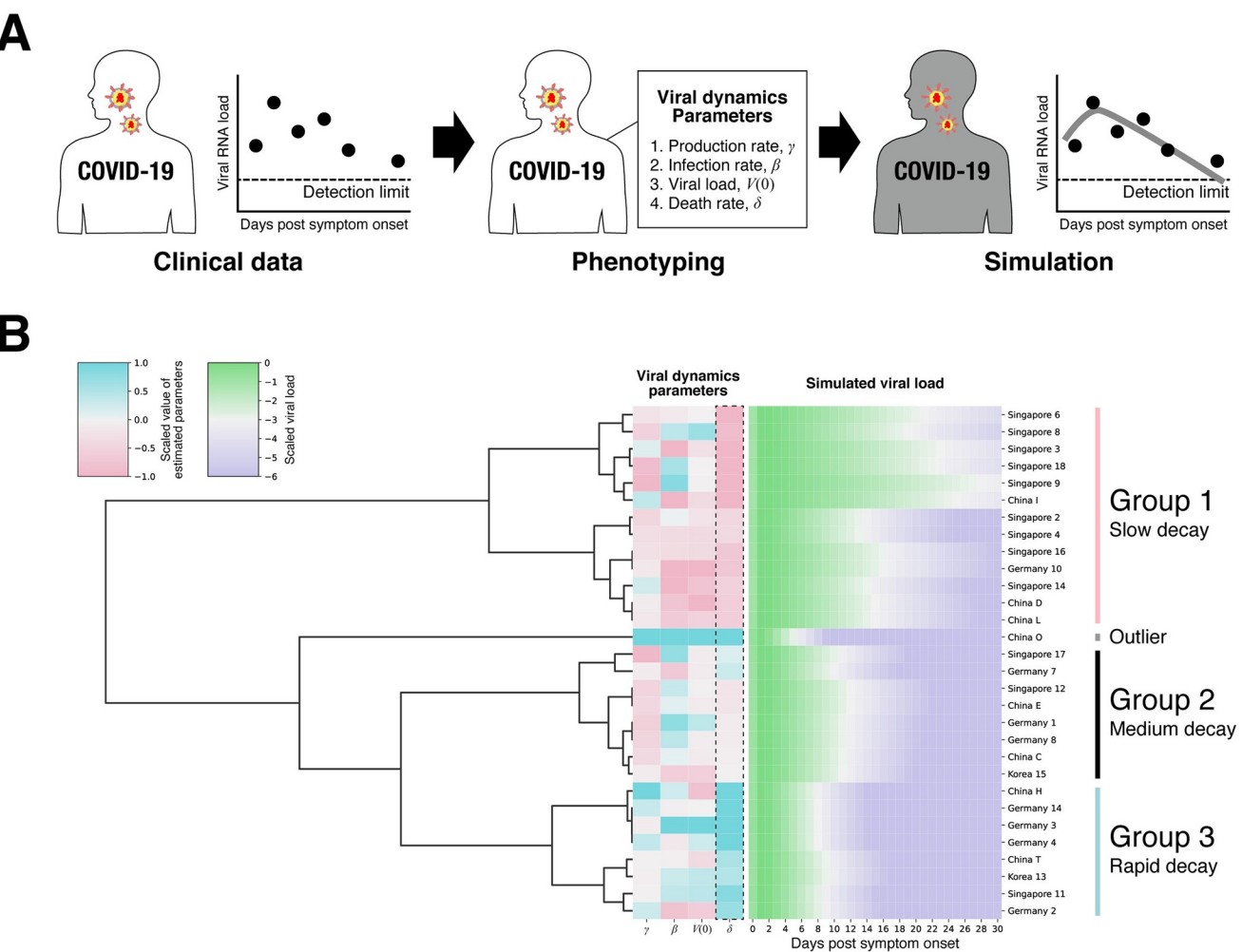

**Fig 2. Characterizing and clustering COVID-19 patients using viral load data.** (A) Schematic illustration for data fitting with a virus dynamics model. Longitudinal SARS-CoV-2 RNA load data (i.e., clinical data) were extracted from published papers. The data were analyzed by the mathematical model, and then virus dynamics parameters were estimated for each patient (i.e., phenotyping). Daily viral load since symptom onset for each patient was simulated by running the model with the estimated parameters. (B) Clustering patients using daily viral load. Daily viral load obtained through simulation was used for clustering of the 30 patients. In the dendrogram, the height from the bottom to the point where 2 or more patients are joined indicates the distance (i.e., dissimilarity) between patients. For example, "Singapore 11" and "Germany 2" are very close, and those are far from "Singapore 6." As a result, 3 different patient groups were identified, and "China O" was detected as an outlier. The heatmap next to the dendrogram ("Virus dynamics parameters") shows the estimated parameters and initial condition ($\gamma$, $\beta$, $V(0)$, $\delta$) for each patient. Light blue and pink correspond to high and low values, respectively. Statistically significant between-group differences were found in the maximum rate constant for viral replication, $\gamma$ (ANOVA $p$-value: $3.61 \times 10^{-3}$), and the death rate of virus-producing cells, $\delta$ (ANOVA $p$-value: $3.23 \times 10^{-13}$); moreover, there were statistical differences between all pair groups for $\delta$. The death rate is highlighted by the dotted square. The right heatmap shows the daily viral load for each patient. Green and purple correspond to high and low values, respectively. "Group 1" maintained a high viral load for a longer period compared with the other groups. The underlying data for this figure can be found in S2 Data. COVID-19, Coronavirus Disease 2019; SARS-CoV-2, Severe Acute Respiratory Syndrome Coronavirus 2.

To understand the source of the difference in virus dynamics between groups, we tested the differences in the estimated parameters (i.e., $\beta$, $\gamma$, $\delta$, and $V(0)$) among the groups. Statistically significant between-group differences were found in the maximum rate constant for viral replication, $\gamma$, and in the death rate of virus-producing cells per day, $\delta$ (S1 Fig). The differences in $\gamma$ are related to the growth of viral load (a larger $\gamma$ indicates more rapid growth); however, the difference in $\gamma$ between groups was sufficiently small that its influence on virus dynamics especially after the viral load peak (or symptom onset) is negligible. The difference in $\delta$ manifests in the speed of viral load decay, i.e., a small value of $\delta$ corresponds to a slow decay in viral load

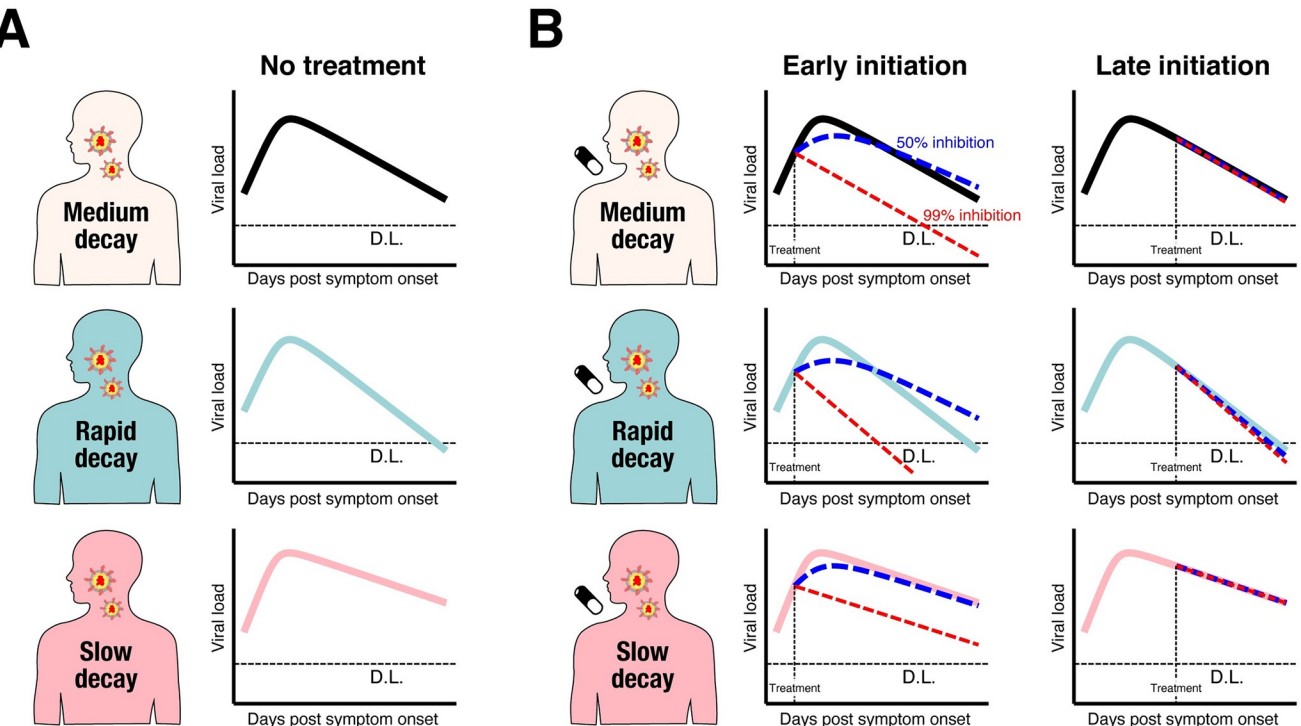

**Fig 3. Patient variability and difference in therapeutic response. (A)** Viral load trajectories since symptom onset for the 3 groups (black: medium decay group, light blue: rapid decay group, and pink: slow decay group) obtained through simulation (without antiviral treatment). **(B)** Viral load trajectories since symptom onset for the 3 groups under antiviral treatment with different inhibition rates and different timing of treatment initiation. The left 3 panels are viral load trajectories when the treatment is initiated at 0.5 days ("Early initiation") since symptom onset. The right 3 panels are vial load trajectories when treatment is initiated at 5 days ("Late initiation") since symptom onset. Blue and red dotted lines correspond to the trajectories with 50% and 99% inhibition rate, respectively. The bolded lines are the trajectory without treatment shown for comparison. The dotted horizontal lines are the D.L. D.L., detection limit.

(Fig 2B). Thus, we named the 3 groups as rapid, medium, and slow viral load decay groups (Fig 3A). The mean decay rates were 1.17 d$^{-1}$ (95% CI: 1.06 to 1.27 d$^{-1}$), 0.777 d$^{-1}$ (0.716 to 0.838 d$^{-1}$), and 0.450 d$^{-1}$ (0.378 to 0.522 d$^{-1}$) for the 3 groups, respectively. The minimum and maximum of the decay rates for the 3 identified groups were 0.270 d$^{-1}$ to 0.616 d$^{-1}$ (slow), 0.700 d$^{-1}$ to 0.914 d$^{-1}$ (medium), and 0.993 d$^{-1}$ and 1.30 d$^{-1}$ (rapid), respectively. The border value of the decay rate between groups is defined as the mean of the highest value in the lower group and the lowest value in the higher group. Thus, the border value of the slow and medium groups was 0.658 d$^{-1}$ [(0.616 + 0.700)/2] and that of the medium and rapid groups was 0.953 d$^{-1}$ [= (0.914 + 0.993)/2].

## SARS-CoV-2 virus dynamics and antiviral effect

Using our mathematical model and the estimated parameter distribution for each patient (S3 Table), we conducted in silico experiments to determine the possible therapeutic response, measured in terms of virus dynamics, of drug treatments blocking virus replication. Clinical outcomes are known to be related to the timing of initiation of antiviral treatment in general, and, especially, for influenza [31,39–42], and the antiviral effects of a treatment are dependent on dose and the patients' immune system [43,44]. Thus, we studied several different scenarios in which we varied the time of treatment initiation (0.5 or 5 days from symptom onset, which were before and generally after the estimated peak viral load in our dataset) and

the inhibition rate (99% or 50%). We resampled a total of 1,000 parameter sets from the estimated parameter distributions for this simulation and separated the individuals according to the value of the viral load decay rate (i.e., rapid, $\delta > 0.953$ d$^{-1}$; medium, $0.658$ d$^{-1} \le \delta \le 0.953$ d$^{-1}$; or slow, $\delta < 0.658$ d$^{-1}$). We found that early initiation of antiviral treatment with a high inhibition rate (i.e., 99%) immediately reduced the viral load after initiation (Fig 2B, S3 Fig). However, if the inhibition rate was low (i.e., 50%), the viral load kept increasing, and the viral load decay rate after the peak was slower or equivalent to that without treatment. This was because viral replication was not efficiently inhibited and thus it continued albeit with a lower rate even after treatment initiation and continued long after the peak. In contrast, virus dynamics was not much influenced if treatment was initiated after the peak regardless of the inhibition rate or the patient type (Fig 3B, S2 Fig), because the number of uninfected targeted cells remaining at this stage of infection is limited. It is intriguing that a weak antiviral effect was observed for patients with rapid decay even when the treatment was initiated after the peak. Because the virus is removed rapidly during the course of infection for patients with rapid decay, more uninfected cells remain compared with the other groups. Therefore, antiviral drugs can mitigate replication of the virus even when treatment is initiated after the peak to some extent. Note that these findings are not unique to SARS-CoV-2; similar findings for virus dynamics and antiviral effects have been suggested in other infectious diseases [25,45].

## Observational studies for antiviral drugs cannot yield significant results owing to heterogeneity in virus dynamics

We explored why compassionate use programs do not yield significant findings when using the duration of virus shedding as an outcome. Duration of virus shedding is one of the most frequently used outcomes for assessing antiviral treatment for SARS-CoV-2 infection (S1 Table) [12,13,15,45]. The distribution of the duration of virus shedding without treatment in the different virus decay groups is shown in Fig 4A and S3A Fig. As can be expected from the difference in viral load dynamics, the duration of virus shedding without treatment is longer in the group with slow viral load decay: The averages in the groups with medium, rapid, and slow decay were 12.3 (SD: 1.06), 8.86 (SD: 1.40), and 22.5 (SD: 8.15) days, respectively. As a sensitivity analysis, we also computed and compared the cumulative viral load (AUC). We confirmed the same trend in the cumulative viral load in log scale (S3B and S4A Figs). We further compared the outcomes under antiviral treatment (inhibition rate was set as 50% and 99%). Regardless of viral decay rate group, we consistently observed that both outcomes were improved by early treatment initiation (day 0.5) but not by late treatment initiation (day 5), as illustrated in Fig 4B and S4B Fig.

If a patient possesses strong viral defenses, including immune-mediated defenses, the virus-producing cells are removed quickly, which corresponds to a shorter duration of virus production and rapid viral load decay. Indeed, the duration of virus shedding in respiratory samples has been associated with disease severity [46] and differs between symptomatic and asymptomatic cases [47]. Taken together, these findings suggest that both treatment allocation and clinical outcomes in compassionate use programs are associated with severity; thus, severity is a potential confounding variable. Further, there may be other confounding variables in the assessment of treatment efficacy in compassionate use programs; however, controlling all of them in the analysis is not possible. For example, heterogeneous immune responses, which are partially represented by the death rate of infected cells in the model, can confound the inference. However, quantifying the immune response is difficult. We need to be careful when interpreting the results from compassionate use programs.

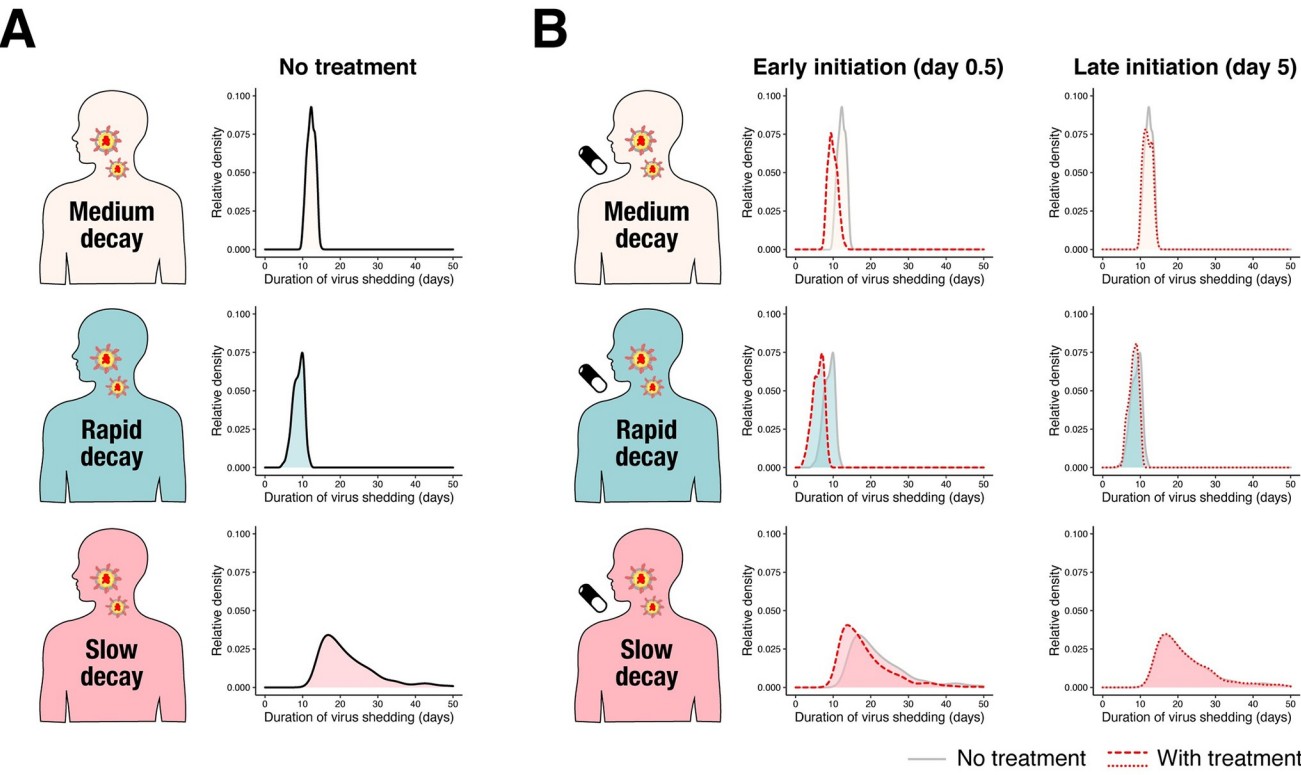

**Fig 4. Duration of virus shedding in the 3 different groups. (A)** The relative density distributions of duration of virus shedding since symptom onset for the 3 groups (light pink: medium decay group, light blue: rapid decay group, and pink: slow decay group) without treatment obtained through simulation. **(B)** The relative density distributions of duration of virus shedding since symptom onset for the 3 groups under antiviral treatment with different inhibition rates and different timing of treatment initiation. The left 3 panels are when antiviral treatment is initiated at 0.5 days ("Early initiation") since symptom onset. The red dotted line corresponds to the distribution with a 99% inhibition rate. The distribution without treatment is shown in the back for comparison. The right 3 panels are when antiviral treatment is initiated at 5 days ("Late initiation") since symptom onset. The distributions are represented as "relative density" to reflect different proportions of the 3 groups. The underlying data for this figure can be found in S3 Data.

## Randomized controlled trials need to enroll patients early after symptom onset to observe significant antiviral effects

In contrast to observational studies, randomized controlled trials may not be influenced by confounding variables and could provide valid inference. However, clinical trials for COVID-19 should consider the timing of treatment initiation in the design (i.e., inclusion–exclusion criteria), because differences in outcomes are unlikely to be observed under late treatment initiation as we demonstrated in the previous section (Fig 4B, S4B Fig).

We computed the sample size needed to observe a statistically significant difference in outcomes with 80% power and a significance level of 0.05 assuming patients are randomly assigned and treated (with antiviral or placebo) immediately after hospitalization with different antiviral effect (95% and 99%) (Fig 5A) and with different inclusion and exclusion criteria for the timing of enrollment. We primarily used the duration of virus shedding as an outcome, but the results are qualitatively consistent for cumulative viral load as an outcome. The sample size is strongly dependent on the criterion of the timing of enrollment, i.e., the sample size can be reduced if patients are enrolled early after symptom onset (S4 Table). The distribution of duration of virus shedding under the different criteria is shown in Fig 5B and 5C. If patients are enrolled regardless of the time of treatment initiation, the sample sizes are 13,603 and 11,670 per group when the inhibition rate is 95% and 99%, respectively, which is much larger

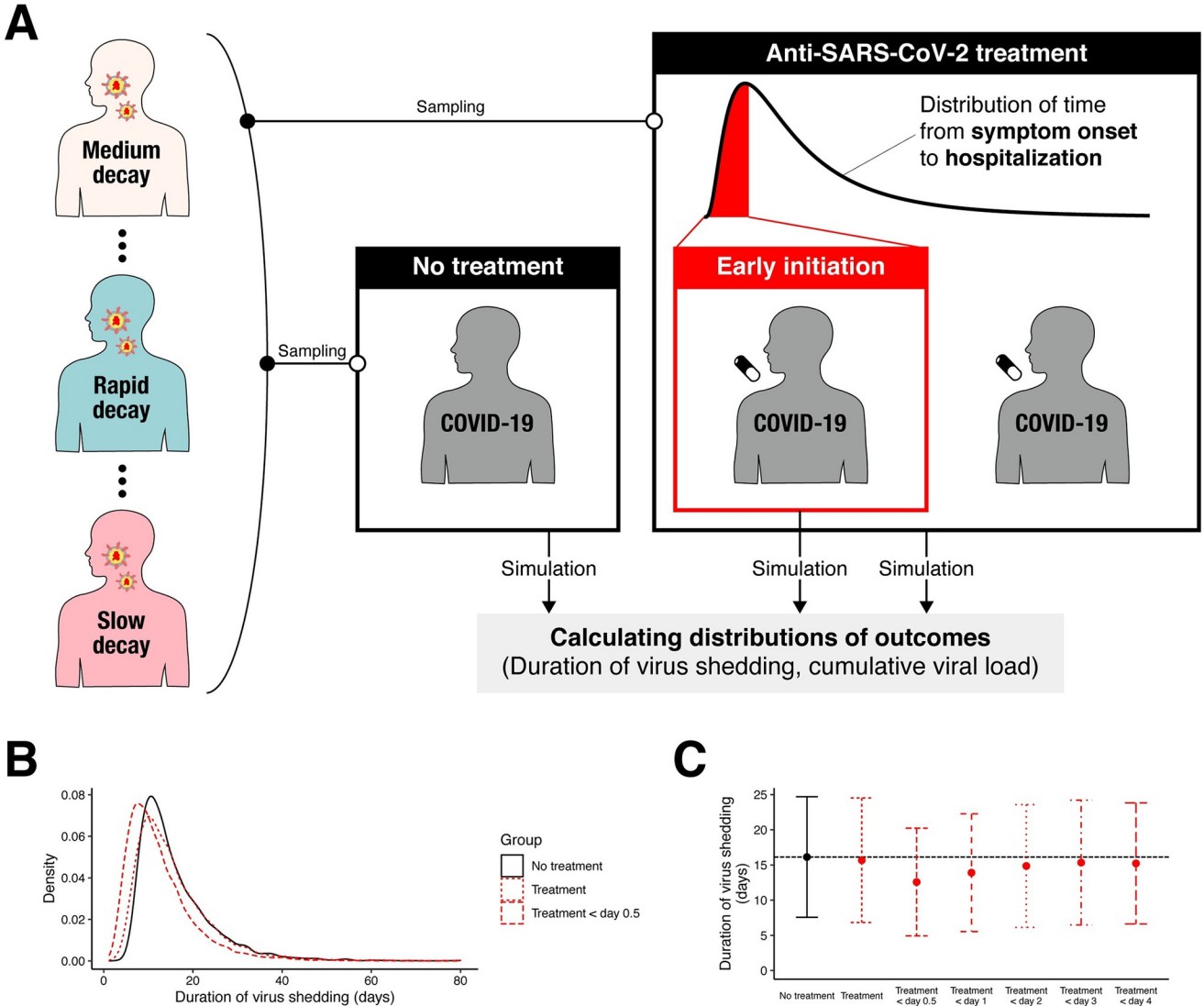

**Fig 5. Simulation mimicking randomized controlled trial for anti-SARS-CoV-2. (A)** Schematic illustration for the simulation mimicking randomized controlled trials for antiviral drugs. A total of 20,000 parameter sets were sampled from the estimated parameter distributions. The parameter sets were randomized into control ("No treatment," 10,000 individuals in total) or treatment ("Anti-SARS-CoV-2 treatment," 10,000 individuals in total) groups. For the treatment group, the treatment is initiated randomly to reflect the delay of treatment initiation since symptom onset. We have also used a truncated distribution (red area) to mimic the randomized controlled trials including only patients recruited and treated early ("Early initiation"). Then the outcomes (duration of virus shedding and the cumulative viral load in log scale) were calculated for each patient. **(B)** The distributions of duration of virus shedding for patients without and with treatment are shown in black and red curves, respectively. For antiviral treatment, a 99% inhibition rate of virus replication was assumed. The red dotted curve is when all patients are included regardless of the timing of recruitment and treatment initiation. The red dashed curve is the case when only patients recruited and treated within 0.5 days since symptom onset are included. **(C)** Means and standard deviation of the duration of virus shedding under different inclusion criteria (not treated, included regardless of the timing of recruitment and treatment initiation, and early treatment initiation [within 0.5, 1, 2, 3, and 4 days since symptom onset]) are shown. The underlying data for this figure can be found in S4 Data. COVID-19, Coronavirus Disease 2019; SARS-CoV-2, Severe Acute Respiratory Syndrome Coronavirus 2.

than the empirical sample size of the randomized controlled trials of antivirals for SARS-CoV-2 (S1 Table). This large sample size is needed given that the treatment is initiated 4.6 days after onset of symptoms on average, which is after the viral load peak. If we enroll only the patients treated within 1 day of the onset of symptoms, the sample size is reduced to 584 and 458 per group when inhibition rate is 95% and 99%, respectively. Note that antiviral drugs with a

weaker inhibition rate will require larger sample sizes. The trend was similar when cumulative viral load in log scale was used as an outcome (S4 Table, S4C and S4D Fig).

## Timing of randomization of clinical studies of antiviral drugs for SARS-CoV-2

To validate our findings from a practical perspective, we checked the clinical trials investigating antiviral efficacy registered in ClinicalTrials.gov. As of May 22, 2020, we identified 176 clinical trials with the search terms "antiviral" and "COVID." Among them, 46 studies did not investigate the efficacy of antiviral drugs (the effect of anti-inflammatory drugs were investigated, for example), and 20 studies did not directly investigate the efficacy of antivirals (such as vaccine studies and safety studies). Among the remaining 110 studies investigating antiviral effect, including remdesivir, chloroquine, and lopinavir/ritonavir, only 17 studies (15%) explicitly stated the time from symptom onset in the inclusion or exclusion criteria. The average time from symptom onset to randomization was 7.2 days, which our findings suggest is too late to observe a statistically significant antiviral effect with a reasonable sample size.

## Discussion

We explored the mechanism behind the inconsistent or null findings of clinical studies of the antiviral effect of treatments for SARS-CoV-2 infection. By fitting a conventional virus dynamics model to the longitudinal viral load data from patients with COVID-19 (without antiviral treatment), we found that there is large heterogeneity in virus dynamics, as characterized by different virus decay rates. Such heterogeneity in virus dynamics could be a confounding factor in observational studies. Subsequently, a set of randomized controlled trials was mimicked by using a version of the model with an antiviral effect. We assumed that therapy was initiated as soon as a participant was hospitalized with COVID-19 symptoms. We used a reported distribution of time delays from symptom onset to hospitalization in China to make the simulation more realistic. When we included all patients in the trial regardless of the timing of randomization and treatment initiation (1:1 allocation for treatment:placebo), we found that more than 11,000 patients per group would need to be recruited. By including only patients hospitalized within 1 day since symptom onset, the sample size is reduced to about 450 per group. Thus, we conclude that clinical trials should consider the time of treatment initiation in the study design.

In randomized controlled trials, the calculation of sample size has been performed directly by assuming specific distributions for outcomes with a prespecified effect size [48]. However, as we demonstrated, the antiviral effect is determined not only by dose and type of drug, but also by the timing of treatment initiation and the parameters that govern the virus dynamics. Further, the association between treatment initiation and the outcome (length of viral shedding) is nonlinear; thus, our mathematical model-based approach can provide a more reliable sample size than a conventional effect size-based approach.

We used measurements related to viral load as outcomes in this study rather than mortality. Mortality is an important and ultimate clinical outcome at both the individual and the population level. However, that does not undermine the value of outcomes related to viral load (such as the duration of virus shedding), which have a different interpretation than clinical outcome. One thing that can be captured by viral load but not by clinical outcome is potential transmissibility. From a clinical viewpoint, each drug has its purpose of use. For example, immunosuppressive agents (e.g., dexamethasone) are expected to reduce clinical symptoms and mortality. Meanwhile, the efficacy of antiviral drugs should be evaluated primarily by using viral load. In addition, the major objective of therapy depends on the severity of disease. Lifesaving is the

most important for patients with severe illness. For mild cases, physicians attempt to prevent the condition and spread of infection from getting worse by using drugs with few adverse effects. A primary endpoint should be determined on the basis of the objectives and goals of clinical trials. As most COVID-19 patients have mild to moderate disease, the duration of viral shedding would be more appropriate than mortality as a primary outcome. Indeed, many studies have used the duration of viral shedding as an outcome (S1 Table).

Regarding the association between viral load and clinical outcomes such as mortality and clinical scores, it has been observed in a number of studies that a high viral load at diagnosis is associated with severe clinical outcomes [46,49,50] and increased risk of mortality [51]. The data we used in this study do not contain clinical outcomes; thus, we cannot correlate our results with clinical outcomes. However, assuming that the viral load at diagnosis is close to the viral load at symptom onset, $V(0)$, we did not find a significant difference in $V(0)$ between groups. The groups we identified were characterized by a difference in the death rate of virus-producing cells that was reflected in the virus decay rate. Combining our findings with those from the literature, disease severity might not be associated with a difference in overall viral dynamics. For prognosis purposes, we need to better understand when and which biomarkers including the viral load differentiate between severe and non-severe cases.

The strength and uniqueness of our approach is that we accounted for virus dynamics in the assessment of antiviral effects and sample size calculations. As far as we know, considering the timing of treatment initiation in a conventional approach to sample size calculation is challenging, especially because the outcome is nonlinearly dependent on the timing of treatment initiation. Even if it is technically possible, the data including the timing of treatment initiation would be limited or small. We used clinical data from SARS-CoV-2–infected patients for the simulation. Thus, our numerical results are realistic and directly interpretable for drug development for SARS-CoV-2. In other words, our approach is flexible and can be applied to other antiviral drugs for other diseases by replacing the dataset.

There are several limitations in our approach. First, our within-host virus dynamics model does not fully reflect the detailed physiological processes of virus replication of SARS-CoV-2. For example, our mathematical model assumed target cells are a homogeneous population (i.e., single-target cell compartment). The susceptibility of target cells for SARS-CoV-2 infection is, however, dependent on expression levels of its receptor, angiotensin converting enzyme 2 (ACE2) [52], and, therefore, susceptibility to infection might be heterogeneous (i.e., multi-target cell compartments) even in the same organ. However, the virus dynamics of our model and that of a model with multi-target cell compartments may not differ substantially unless a large fraction of the total target cells in the modified model remain uninfected around peak viral load. Another modeling limitation is that possible immunomodulation induced by treatment was not modeled. That is, if anti-SARS-CoV-2 drugs induce immunomodulation as bystander effects, late initiation of treatments might still have the potential to reduce viral load, which is not reflected in our model [31]. We further compared the results of the model we used in this study with 2 other extended models, which have been used to describe virus dynamics of SARS-CoV-2 and other viruses, to check whether our model is appropriate: one included the effect of interferons produced by infected cells [53] and the other included the eclipse phase of infection [33,53]. We fit these models to the data and used model selection theory to compare the models based on the Bayesian information criteria (BIC) and corrected Bayesian information criteria (BICc). BIC and BICc among the 3 models were comparable. In addition, given limited data (i.e., only viral load data were available), we believe using a minimal model is appropriate at this stage of knowledge. At such time that further data and appropriate scientific information about infection dynamics becomes available, more complex models may be able to capture additional details of within-

host viral dynamics. Second, we did not use viral load data under treatment to evaluate our model because such data were not sufficiently available. Estimating antiviral effects from such data and using that in the sample size calculation would strengthen our approach, but we need to wait until such data are accumulated.

Future study should include development of a similar sample size calculation framework for different types of antiviral treatment. Although we focused on drugs inhibiting virus replication, there are different classes of drugs such as viral entry inhibitors (e.g., hydroxychloroquine and camostat) and immunomodulators (e.g., interferon and the related agents) [54].

Along with vaccines, developing effective antiviral drugs is urgently needed. At present, most of the randomized controlled trials have failed to identify effective antiviral agents against SARS-CoV-2. However, this might not be because the antiviral drugs are not effective, but because of imperfect design of the clinical studies. The timing of treatment initiation and virus dynamics should be accounted for in the study design (i.e., sample size and inclusion–exclusion criteria). We further believe that our approach is informative for determining treatment strategy in clinical settings.

## Supporting information

**S1 Fig. Estimated parameters of the mathematical model in the 3 different viral load decay groups.** Boxplots of estimates of **(A)** the rate constant for virus infection, $\beta$; **(B)** the maximum rate constant for viral replication, $\gamma$; **(C)** the death rate of virus-producing cells, $\delta$; and **(D)** viral load at symptom onset, $V(0)$, respectively. Estimated parameter distributions between the 3 groups with different viral load dynamics (slow, medium, and rapid viral load decay groups) were compared by ANOVA. Pairwise comparison was subsequently performed using Student $t$ test. The $p$-values of the pairwise Student $t$ test were adjusted by the Bonferroni correction. The underlying data for this figure can be found in S2 Table.
(TIF)

**S2 Fig. Expected virus dynamics under an antiviral treatment blocking viral replication.** The antiviral treatment was assumed to be initiated after 0.5 or 5 days (named early and late initiations, respectively) from symptom onset with 99% and 50% inhibition rates (named high and low antiviral effects, respectively) for patients with **(A)** medium, **(B)** rapid, and **(C)** slow viral load decay. Left and right panels in each group show the viral loads, $V(t)$, and the relative fraction of uninfected target cells, $f(t)$. The black and colored solid lines correspond to the mean of the values without and with the therapies (red: high, blue: low antiviral effects), respectively. The shadowed regions correspond to the 95% predictive intervals. The underlying data for this figure can be found in S5 Data.
(TIF)

**S3 Fig. Distribution of duration of virus shedding and cumulative viral load. (A)** Duration of virus shedding and **(B)** cumulative viral load of SARS-CoV-2 for different types of patients (medium, rapid, and slow decay) with and without treatment. "Early initiation" and "Late initiation" mean the early and the late treatment initiation (0.5 or 5 days after symptom onset). The dots and error bars represent the mean and the standard deviation. The underlying data for this figure can be found in S3 Data. SARS-CoV-2, Severe Acute Respiratory Syndrome Coronavirus 2.
(TIF)

**S4 Fig. Cumulative viral load in the 3 different groups. (A)** The relative density distributions of cumulative viral load for the 3 groups (light pink: medium decay group, light blue: rapid decay group, and pink: slow decay group) without treatment obtained through

simulation. **(B)** The relative density distributions of cumulative viral load (in log scale) for the 3 groups under antiviral treatment with different inhibition rate and different timing of treatment initiation. The left 3 panels are when antiviral treatment is initiated at 0.5 days ("Early initiation") since symptom onset. The red dotted line corresponds to the distribution with 99% inhibition rate. The distribution without treatment is shown in the back for comparison. The right 3 panels are when antiviral treatment is initiated at 5 days ("Late initiation") since symptom onset. The distributions are represented as "relative density" to reflect different proportions of the 3 groups. **(C)** The distributions of cumulative viral load for patients without and with treatment are shown in black and red curves, respectively. The red dotted curve is when all patients are included regardless of the timing of recruiting and treatment initiation. The red dashed curve is the case for patients who were recruited and who received treatment within 0.5 days since symptom onset only. **(D)** Means and standard deviation of the cumulative viral load under different inclusion criteria (not treated, included regardless of the timing of recruiting and treatment initiation, and early treatment initiation [within 0.5, 1, 2, 3, and 4 days since symptom onset]) are shown. The underlying data for this figure can be found in S3 and S4 Data.
(TIF)

**S1 Table. Summary of clinical trials for antiviral drugs for SARS-CoV-2.** Current major clinical studies for antiviral treatment of SARS-CoV-2 (as of May 22, 2020) were investigated and their information were summarized. SARS-CoV-2, Severe Acute Respiratory Syndrome Coronavirus 2.
(DOCX)

**S2 Table. Estimated parameters for each patient (mode and 95% credible intervals).** The conditional modes of the individual parameters for each patient were estimated as empirical Bayes estimates and summarized. The patient type was based on the 3 groups identified by the hierarchical clustering of the reconstructed daily viral load data.
(DOCX)

**S3 Table. Description of variables, parameters, and estimated parameter values.** The fixed effect and the random effect for each parameter were estimated by the nonlinear mixed-effect model. Estimated values with standard errors were summarized.
(DOCX)

**S4 Table. Sample size (per group) under different inclusion criteria.** Sample size was computed for different outcomes (duration of virus shedding and cumulative viral load) under assumed inhibition rate (95% or 99%) and inclusion criteria (0.5 days to 4 days). The underlying data for this figure can be found in S4 Data.
(DOCX)

**S1 Data. Original numerical values for Fig 1.**
(XLSX)

**S2 Data. Original numerical values for Fig 2.**
(XLSX)

**S3 Data. Original numerical values for Fig 4 and S4A and S4B and S3 Figs.**
(XLSX)

**S4 Data. Original numerical values for Fig 5, S4C and S4D Fig, and S4 Table.**
(XLSX)

**S5 Data. Original numerical values for S2 Fig.**
(XLSX)

## Author Contributions

**Conceptualization:** Keisuke Ejima, Shingo Iwami.

**Formal analysis:** Shoya Iwanami, Keisuke Ejima, Kwang Su Kim, Koji Noshita, Yasuhisa Fujita, Shingo Iwami.

**Funding acquisition:** Shoya Iwanami, Kazuyuki Aihara, Koichi Watashi, Alan S. Perelson, Shingo Iwami.

**Investigation:** Shoya Iwanami, Keisuke Ejima, Kwang Su Kim, Koji Noshita, Yasuhisa Fujita, Shingo Iwami.

**Methodology:** Shoya Iwanami, Keisuke Ejima, Kwang Su Kim, Koji Noshita, Yasuhisa Fujita, Shingo Iwami.

**Project administration:** Keisuke Ejima, Alan S. Perelson, Shingo Iwami.

**Supervision:** Keisuke Ejima, Alan S. Perelson, Shingo Iwami.

**Validation:** Shoya Iwanami, Kwang Su Kim.

**Visualization:** Shoya Iwanami, Kwang Su Kim.

**Writing – original draft:** Shoya Iwanami, Keisuke Ejima, Katsuhito Fujiu, Alan S. Perelson, Shingo Iwami.

**Writing – review & editing:** Shoya Iwanami, Keisuke Ejima, Kwang Su Kim, Yasuhisa Fujita, Taiga Miyazaki, Shigeru Kohno, Yoshitsugu Miyazaki, Shimpei Morimoto, Shinji Nakaoka, Yoshiki Koizumi, Yusuke Asai, Kazuyuki Aihara, Koichi Watashi, Robin N. Thompson, Kenji Shibuya, Katsuhito Fujiu, Alan S. Perelson, Shingo Iwami, Takaji Wakita.

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
