## [Editor Report · Decision Letter 0]

30 Jun 2020

Dear Dr Iwami, 

Thank you for submitting your manuscript entitled "Rethinking antiviral effects for COVID-19 in clinical studies: early initiation is key to successful treatment" for consideration by PLOS Medicine.

Your manuscript has now been evaluated by the PLOS Medicine editorial staff and I am writing to let you know that we would like to send your submission out for external peer review.

Kind regards,

Artur Arikainen,

Associate Editor

PLOS Medicine

---

## [Decision Letter · Decision Letter 1]

29 Jul 2020

Dear Dr. Iwami,

Thank you very much for submitting your manuscript "Rethinking antiviral effects for COVID-19 in clinical studies: early initiation is key to successful treatment" (PMEDICINE-D-20-03078R1) for consideration at PLOS Medicine. 

[LINK]

In light of these reviews, I am afraid that we will not be able to accept the manuscript for publication in the journal in its current form, but we would like to consider a revised version that addresses the reviewers' and editors' comments. Obviously we cannot make any decision about publication until we have seen the revised manuscript and your response, and we plan to seek re-review by one or more of the reviewers. 

We expect to receive your revised manuscript by Aug 19 2020 11:59PM. Please email us (plosmedicine@plos.org) if you have any questions or concerns.

We look forward to receiving your revised manuscript. 

Sincerely,

Artur Arikainen, 

Associate Editor 

PLOS Medicine

plosmedicine.org

1. Please address all of the reviewers’ comments below – further consideration of your manuscript will depend on satisfactory resolution of the outstanding concerns.

2. Please revise your title according to PLOS Medicine's style. Your title must be nondeclarative and not a question. It should begin with main concept if possible. "Effect of" should be used only if causality can be inferred, i.e., for an RCT. Please place the study design ("A randomized controlled trial," "A retrospective study," "A modelling study," etc.) in the subtitle (ie, after a colon).

3. All authors must declare their relevant competing interests per the PLOS policy, which can be seen here: https://journals.plos.org/plosmedicine/s/competing-interests

For authors with ties to industry, please indicate whether any of the interests has a financial stake in the results of the current study. Specifically, please mention employment by authors to Science Groove Inc.

4. Please include line numbers in the margin for all lines, preferably not restarting at 1 on each page.

5. Abstract: 

a. Please quantify all of the main results (with 95% CIs and p values), eg. decay rates, sample sizes.

b. In the last sentence of the Abstract Methods and Findings section, please describe the main limitation(s) of the study's methodology.

7. Please remove spaces from within citation callouts, eg: “...and in vivo models [2,3].”

8. Please present and organize the Discussion as follows: a short, clear summary of the article's findings; what the study adds to existing research and where and why the results may differ from previous research; strengths and limitations of the study; implications and next steps for research, clinical practice, and/or public policy; one-paragraph conclusion.

-----

Comments from the reviewers:

Reviewer #1: Iwanami and colleagues use a mathematical modelling approach to predict the necessary sample size in SARS-CoV-2 clinical trials. This is an interesting question and different approach from the usual. The authors develop a model of viral growth and decline based on published data, and try to predict how much viral replication might be blocked by treatment at different times (and how hard it would be to detect this). 

Major issues:

1) The effectiveness of a drug early versus late assumes different levels of ongoing replication are happening at the time. Therefore, growth to peak, when peak occurs, how much replication after peak etc, are major factors. However, the data are almost exclusively on what happens after peak. Supp Fig 1 (which needs to be in the main figures, as it is the foundation of the whole analysis) shows that growth rate data is available for only a very few patients (Singapore 16, France 4, (who reach their peaks at 10 and 4 days post-symptoms), maybe a few others). So the statement "the peak of viral load appeared 2-3 days from symptom onset, which was consistent for all patients" is very misleading. For most patients, they were at their highest level when first sampled (and peak was anytime before that). Thus, all the viral growth parameters are highly questionable, as is any assumption that earlier treatment would help (since the peak may have even been before symptoms).

2) The model depends on 'target cell limitation' being the only factor determining viral decay. As a result of this assumption,it is assumed that there is very little ongoing infection after the peak (and thus blocking viral replication can have little effect). However, this also assumes that infected cells are very long lived and hardly affected by immune responses. If instead the decay phase is a balance of replication and death, then antiviral treatment may have a much greater effect. However, since as in point 1, nothing meaningful can be said around growth, it is hard to see that any conclusions on viral replication (or its inhibition by treatment) are very reliable.

3) There seems an assumption that clinical trials would mainly depend on duration of viral shedding as a primary outcome. Mortality is a much more definitive and important outcome. This and other clinical outcomes may well be much more useful than viral load. Indeed, since mortality occurs well after presentation (when viral loads are declining in mild infection, but declining more slowly in severe), this suggests that ongoing infection may play a larger role. In any event, studies of clinical outcome may require far fewer patients - so why would one rely on viral load outcomes (which could only ever be useful as a proxy of clinical outcomes)?

4) The major conclusion - that duration of viral shedding is highly variable and thus there is low statistical power to detect differences - is an important one. You don't need a model to conclude this (just analysing the variability in duration of detection would be useful). However, the conclusion from this would be that the proxy measure (viral load) is less useful than the most important measure (clinical outcome)?

Reviewer #2: Statistical review of the manuscript 'Rethinking antiviral effects for COVID-19 in clinical studies: early initiation is key to

successful treatment' by Shoya Iwanami and colleagues, submitted to PLOS Medicine.

Summary

In this manuscript, Iwanami and colleagues extract data from published longitudinal viral load data of COVID-19 (three studies) and fit a simple within-host model to these data. Using these fits (i.e. parameter estimates), the authors use clustering methods to stratify the viral dynamics as fast, intermediate, and slow. Subsequently, the model armed with parameter estimates is used to gauge the impact of antiviral drugs interfering with viral replication. Based on this, the authors make suggestions as to the sample size needed to detect different in a hypothetical clinical trial. 

Evaluation

I like the central idea of the manuscript very much, and believe that approaches such as the one in this manuscript, confronting real-world data with mechanistic within-host models, are highly commendable. HOWEVER, I was considerably less impressed with many details of the study (see below), presentation of the results, and sensitivity analyses. In essence, I would consider the current manuscript an (very interesting) working draft that needs substantial work before it would be suitable for publication. Therefore, I would argue that a rejection with a strong encouragement for resubmission of a thoroughly redrafted manuscript would be appropriate. 

Specific comments

p3, Conclusions. Vague. Try to come up with specific suggestions/insights.

p4, l27, "we admit this is not an exhaustive list". Mention all clinical studies (my preferences, as this is so central to the current ms), or drop.

p4, l36, "truly effective". This requires at the very least mention of an effect size. I found this missing at several other places.

p5l 9, "the detection limit". First, I found the description of the data overly short. Second, and more importantly, it is not clear how these the data below the detection limit have been included. The text now suggests that in the estimation procedure these left-censored values have been included as actual measurements, rather than as left-censored values (using cumulative density functions as is standard practice). Please repair if necessary, or the very least explain in more detail.

p5, "Mathematical model". Here a very simple within-host model with target cell limitation is used. Several other models and choices (e.g. wrt functional responses) are possible (e.g., one of the authors, ASP, is an expert on this), and these should be explored, and formally evaluated wrt ability to fit to the data.

p5, l28-31. The details on virus replication blocking drugs are interesting, but not in the right place in the section on the model.

p5, l34. Is the the only way drugs can act on viral dynamics. I agree that fairly simple models are to be favored over complex ones, but would suggest that at least some mention should be made of other ways drug action can be incorporated.

p6. l5ff. This is much too short and uninformative, to the extent that it is not possible to reconstruct what the authors have done exactly. The information in the SI only partly helps repair this. Please extend the description. In addition, is there an significant effect of study from which the data were taken? Why not employ a fully Bayesian methods, e.g. using JAGS or Stan? This would result in availability of immediate credible intervals, parameter correlations, etc. Now none of the plots have any indication of the levels of uncertainty surrounding estimates and viral kinetics. 

p6, l8. Please mention if/how the results would be affected if no rescaling by maximum values would have been employed. How are the parameter estimates affected? How would the clustering be affected? Does the clustering actually yield a clustering by max height of viral dynamics? Please make the raw data available.

p6, l10. Is the clustering robust to other clustering methods than Ward (e.g., single linkage)? Please check.

p7, l11, "The reconstructed viral loads ...". Actually not the loads but the parameter estimates are analyzed by cluster analysis.

p7, l20-22. Is there a correlation between groups and symptom scores. Please check the original literature whether this information can be retrieved.

p9. Mention of effect size (now only in SI, and assumed to be 99%) is crucial. Also, exploration of sample size with far less effective drugs is warranted!

p10-11. Discussion is a bit superficial, with too many vague/general statements. Try to be precise, and try to single out what makes this study so exciting. 

Reviewer #3: This is a potentially important paper quantifying whether or not the effectivity of novel treatments to COVID-19 can be detected in small groups of patients. By fitting a standard mathematical model to clinical data, the study establishes a marked heterogeneity the viral dynamics among the patients. This study provides suggestions for improvement that would help clinicians to sooner establish the effectivity of treatments.

That being said, the manuscript is written in a rather poor style, and this sometimes reduces the clarity of the text. My major recommendation is therefore to markedly improve the text linguistically.

Secondly, the figures in the main text have no parameter values in their legends, and they contain a lot of useless redundancy (large human figures, no treatment in Panel A is repeated in B).

[LINK]

---

## [Decision Letter · Decision Letter 2]

3 Feb 2021

Dear Dr. Iwami,

Thank you very much for re-submitting your manuscript "Detection of significant antiviral drug effects on COVID-19 with reasonable sample sizes in randomized controlled trials: a modeling study combined with clinical data" (PMEDICINE-D-20-03078R2) for review by PLOS Medicine.

I have discussed the paper with my colleagues and the academic editor and it was also seen again by Reviewer 3 and Reviewer 4. I am pleased to say that provided the remaining comments by Reviewer 4 are dealt with we are planning to accept the paper for publication in the journal.

[LINK]

We look forward to receiving the revised manuscript by Feb 10 2021 11:59PM.   

Sincerely,

Dr Raffaella Bosurgi MSc, PhD

Executive Editor, PLOS Medicine

rbosurgi@plos.org

https://twitter.com/raffi74

Remote based in London, UK

PLOS

Requests from Editors:

Comments from Reviewers:

Reviewer #3: Thank for improving the manuscript. My previous concerns have all been addressed.

Reviewer #4: In this manuscript, the authors focused on the factors which might affect the power of clinical studies for antiviral drugs against SARS-CoV-2. Longitudinal viral load data from three published studies were extracted and fitted in a virus dynamics model. They report two essential factors in clinical studies: heterogeneity in virus dynamics among patients and late timing of treatment initiation. If patients could be early recruited in randomized controlled trials, a more significant outcome would be observed even with a sample size. Above all, it is a quite interesting approach with important practical significance. Maybe the model is overly simplistic to mimic the viral growth kinetics after treatment, since there are many interactional factors, such as the exposure dose, colonization, genetic background, immunoregulation, and so on. In general, the later the treatment is introduced, the more factors will show their impact on viral replication, leading to a more confounding outcome. However, this study revealed the key association between the timing of recruiting and viral shedding/ cumulative viral load. 

Almost all issues made by previous reviewers have been well solved. However, one issue still needs to be addressed: the viral load at the time of diagnosis. Many adverse outcomes were observed in patients with high viral loads at the initiation of treatment. Several estimated parameters, including , , , and (0), were applied, while the role of viral load at (0) should be evaluated or explained.

[LINK]

---

## [Decision Letter · Decision Letter 3]

12 May 2021

Dear Dr. Iwami,

Thank you very much for re-submitting your manuscript "Detection of significant antiviral drug effects on COVID-19 with reasonable sample sizes in randomized controlled trials: a modeling study combined with clinical data" (PMEDICINE-D-20-03078R3) for review by PLOS Medicine.

I have discussed the paper with my colleagues and the academic editor. I am pleased to say that provided the remaining editorial and production issues are dealt with we are planning to accept the paper for publication in the journal.

[LINK]

We look forward to receiving the revised manuscript by May 19 2021 11:59PM.   

Sincerely,

Louise Gaynor-Brook, MBBS PhD

Associate Editor 

PLOS Medicine

plosmedicine.org

Requests from Editors:

Abstract:

Abstract Background: The final sentence should clearly state the study question.

Line 62 - please amend to ‘my fail to’

Abstract Methods and Findings:

Please ensure that it is made clear with the abstract that this is a modelling study.

Please ensure that all numbers presented in the abstract are identical to numbers presented in the main manuscript text - there seems to be some differences in the decay rates presented in the abstract and results (line 300)

In the last sentence of the Abstract Methods and Findings section, please describe the main limitation(s) of the study's methodology.

Abstract Conclusions:

Please begin your conclusions with "In this study, we observed ..." or similar, to summarise your main findings, and go on to emphasize what is new.

Author Summary:

Within the ‘Why was this study done?’ section, please clearly state the study question.

Introduction:

Line 161 - please temper assertions of primacy by adding ‘to the best of our knowledge’ or similar

Methods:

Line 269 - please define AUC at first use

Results: 

Line 300 - Please ensure that all numbers presented in the abstract are present and identical to numbers presented in the main manuscript text. Please also provide 95% CI as you have for the data in the abstract.

References:

Please ensure that journal name abbreviations match those found in the National Center for Biotechnology Information (NCBI) databases, and are appropriately formatted and capitalised. 

Please remove extra information contained within reference 39

Supplementary files: 

Please provide legends for each individual table in the Supporting Information.

Please include information on the date on which the summary of clinical trials for antiviral drugs for SARS-CoV-2 was completed. 

[LINK]

---

## [Editor Report · Decision Letter 4]

18 May 2021

Dear Dr Iwami, 

On behalf of my colleagues and the Academic Editor, Mirjam E. E. Kretzschmar, I am pleased to inform you that we have agreed to publish your manuscript "Detection of significant antiviral drug effects on COVID-19 with reasonable sample sizes in randomized controlled trials: a modeling study combined with clinical data" (PMEDICINE-D-20-03078R4) in PLOS Medicine.

PRESS

Sincerely, 

Louise Gaynor-Brook, MBBS PhD 

Associate Editor 

PLOS Medicine